# Chart2Csv: Can VLMs Faithfully Convert Complex Charts into Structured Tables?

## Abstract

Charts are widely used for visualizing data in research findings, and many applications require extracting data from charts and converting it into structured tables for large-scale processing and analysis. While vision-language models (VLMs) have shown promising results on chart digitization and understanding tasks, their effectiveness in fully automating this process remains unclear. Existing benchmarks fall short because (1) they contain overly simplified charts that do not reflect real-world complexity, (2) they fail to comprehensively evaluate critical model capabilities, including perception, reasoning, planning, and long-form output generation, and (3) they lack evaluations on both the completeness and accuracy of the structured outputs. To systematically evaluate the performance of VLMs in extracting and structuring data from charts, we introduce Chart2Csv, a benchmark comprising 812 charts sourced from research papers across 5 scientific domains, paired with expert-validated ground-truth CSVs. In Chart2Csv, VLMs are tasked with extracting data points from these charts and converting them into CSVs. We evaluate 16 VLMs on Chart2Csv and find that even the best-performing model, Claude 3.5 Sonnet, misinterprets nearly half of the data points, underscoring the deficiency of existing VLMs in automating chart data extraction and structuring.

## 1 Introduction

Charts are commonly used to visualize data in research findings, and a wide range of applications require extracting data points from these charts in the absence of the original raw data and transforming them into machine-readable formats (e.g., CSV files). For example, reproduction packages for social science research often include only code for reproducing results, omitting the code to recreate visualizations (i.e., the plots and tables) using the original results. Thus, researchers must extract data from original publications to compare with reproduced results (Brodeur et al., 2024; Collaboration, 2012; 2015). Similarly, crucial insights into climate change (Carey, 2012) and ecological evolution (Pagel, 1999; Nundloll et al., 2022) can be obtained from data recorded as charts in historical, scanned, or even hand-written documents. Furthermore, extracting data from charts plays a vital role in enhancing the accessibility of research findings for blind and visually impaired individuals (Mishra et al., 2022).

Traditional computer vision (CV) tools and libraries struggle to recognize numerical values, detect structural elements, and interpret chart components (Figure 1; Appendix A.1). While vision-language models (VLMs) achieve stronger performance on general visual tasks, they still exhibit critical limitations in automating the extraction and structuring of data from charts. As illustrated in Figure 1, even state-of-the-art VLMs (OpenAI, 2024a; Qwen, 2024a; Google, 2024b; Anthropic, 2024; 2025) struggle with accurately extracting data points from charts (Figures 1b and 1c) and comprehensively structuring them into well-formatted tables (Figures 1a and 1d).

To systematically evaluate and highlight the limitations of VLMs in extracting the full set of data points from charts and transforming them into structured tables, we first examine existing benchmarks and summarize their scope in Table 1. These benchmarks, including those for chart digitization and chart understanding, fall short of capturing the full requirements of real-world applications. *First, context-wise,* the input charts used in existing benchmarks are overly simplified and do not accurately reflect the complexity of real-world scenarios. Notably, none of these benchmarks include charts that have been reproduced and validated by human experts, which is a critical aspect

(a) Table 3 of Allcott et al. (2022)

(b) Figure 1 of Ono & Zilis (2022)

(c) Figure 3 of Gsottbauer et al. (2022)

(d) Figure 1 of Wilson (2022)

Figure 1: Example charts in CHART2CSV and the corresponding model performance deficiencies. For all traditional CV libraries (labeled with rectangular frames), we examine their deficiencies in local data extraction, as they are unable to directly generate valid CSV outputs.

Table 1: Features of chart benchmarks.

| Name | Data | | | Task | | | | Metric |
| | Research Findings | Numeric Annot. Removed | Verified | (Exhaustive) Perception | Reasoning | Planning | Complex Outputs | Chart-level Performance |
|---|---|---|---|---|---|---|---|---|
| *Digitalization* | | | | | | | | |
| FigureQA (Kahou et al., 2018) | ✗ | ✗ | ✗ | ✓ | ✗ | ✗ | ✗ | ✗ |
| ExcelChart400K (Luo et al., 2021) | ✗ | ✗ | ✗ | ✓ | ✗ | ✗ | ✓ | ✗ |
| WebData (Choi et al., 2019) | ✗ | ✗ | ✗ | ✓ | ✗ | ✗ | ✓ | ✗ |
| *Understanding* | | | | | | | | |
| CharXiv (Wang et al., 2024) | ✓ | ✓ | ✗ | ✗ | ✓ | ✗ | ✗ | ✗ |
| ChartBench (Xu et al., 2024) | ✗ | ✓ | ✗ | ✗ | ✗ | ✗ | ✗ | ✗ |
| ChartQA (Masry et al., 2022) | ✗ | ✗ | ✗ | ✗ | ✓ | ✗ | ✗ | ✗ |
| MathVista (Lu et al., 2024) | ✗ | ✓ | ✗ | ✗ | ✓ | ✓ | ✗ | ✗ |
| DVQA (Kafle et al., 2018) | ✗ | ✓ | ✗ | ✗ | ✓ | ✗ | ✗ | ✗ |
| Table-LlaVa (Zheng et al., 2024) | ✗ | ✗ | ✗ | ✗ | ✓ | ✗ | ✗ | ✗ |
| **CHART2CSV** | ✓ | ✓ | ✓ | ✓ | ✓ | ✓ | ✓ | ✓ |

for ensuring the relevance and authenticity of the task. *Second, model capabilities-wise,* successfully performing this task requires that VLMs incorporate: (1) perceptual abilities to accurately extract values from visual representations, (2) reasoning skills to comprehend relationships among chart components, (3) planning capabilities to correctly map extracted values to their corresponding positions in structured tables, and (4) aptitude for generating long-form outputs that include all extracted data points in CSV format. Existing benchmarks fail to assess these four capabilities concurrently. *Third, evaluation-wise,* none of the existing benchmarks provide metrics to quantify VLM performance at the chart level, while evaluating model performance in terms of both accuracy and completeness is essential for investigating their applicability. We illustrate the critical gaps of existing benchmarks with detailed examples in Section 2.

To overcome these limitations, we create CHART2CSV, a benchmark designed to evaluate the capability of VLMs in extracting data points from charts in research findings and transforming them into structured tables. We select 275 charts containing plots and 537 charts containing tables from research papers across 5 domains. We manually annotate a ground-truth CSV for each chart. To do this, we apply traditional CV libraries for preliminary data extraction and resolve discrepancies

before organizing the results into structured ground truths. For each task, VLMs take a chart as input and are tasked to generate CSV-formatted outputs containing all the data in the chart. We introduce the details of CHART2CSV in Section 3.

We evaluate the performance of 16 VLMs with promising results across a wide range of benchmarks using various prompting techniques on CHART2CSV. For evaluation, we use overall accuracy as the metric for tables and overall precision as the metric for plots. Claude 3.5 Sonnet achieves the highest accuracy of 0.51 on table charts and the highest precision of 0.51 on plot charts. Our empirical analysis indicates that existing VLMs perform poorly on CHART2CSV tasks, where even the best-performing model misinterprets nearly half of the data points, highlighting their limitations in extracting and structuring chart data. We present the detailed experiment setup in Section 4, and a comprehensive analysis of the results in Section 5.

## 2 LITERATURE REVIEW

We review existing chart benchmarks that (1) evaluate VLM capabilities to extract data from charts (i.e., *chart digitalization*) (Luo et al., 2021; Choi et al., 2019; Kahou et al., 2018) and (2) assess VLM capabilities to understand charts (i.e., *chart understanding*) (Wang et al., 2024; Xu et al., 2024; Masry et al., 2022; Lu et al., 2024; Kafle et al., 2018; Zheng et al., 2024). We recognize limitations from the following three aspects.

First, existing chart benchmarks include overly simplified contexts. Existing chart digitalization benchmarks (Luo et al., 2021; Choi et al., 2019) focus on simple charts in standard forms with clear annotations, whereas real-world charts can be far more complex, as we demonstrate in Figure 1. Existing chart understanding benchmarks include question-answering tasks that demand either a focused examination of specific chart segments for detailed information (e.g., "For the subplot at row 2 and column 1, do any lines intersect?" ($Q_1$) (Wang et al., 2024)) or a general comprehension of the entire chart (e.g., "How many bars are compared?" ($Q_2$) (Kafle et al., 2018)). However, when converting a chart into a structured table, VLMs should closely examine all fine details throughout the entire chart. Moreover, a significant portion of these benchmarks focus solely on tables (Zheng et al., 2024), which compared to plots, lack the visual complexity, spatial layout, and multi-modal reasoning intergation that are central to real-world chart interpretation tasks.

Second, the task requirements of existing VLM benchmarks limit the complexity and scope of model responses. Existing chart digitalization benchmarks either require outputting only the critical components of charts, such as the bounding boxes of bar charts (Luo et al., 2021), or separate the extraction of texts from data values (Choi et al., 2019). Existing chart understanding benchmarks require only simple textual responses. For example, answering $Q_1$ requires generating a single token "Yes" or "No", and answering $Q_2$ requires generating a single digit. However, tasks involving comprehensive data point extraction and structuring necessitate that VLMs generate outputs in complex formats and of extended lengths to effectively represent the extracted data.

Third, existing benchmarks lack rigorous evaluation metrics that emphasize both the completeness and accuracy of the final structured table outputs. Existing chart digitalization benchmarks (Luo et al., 2021; Choi et al., 2019) do not assess the generated table as a whole, focusing instead on partial or element-level recognition. Existing chart understanding benchmarks priortize reasoning tasks over numeric fidelity. For example, "According to this chart, at Month 5, the visitors of Platform A is higher than Platform B." (Xu et al., 2024) evaluates whether VLMs can correctly compare the relative values of Platforms A and B, while converting this chart to a structured table require the precise extraction of the exact values for both platforms.

## 3 CHART2CSV

We create CHART2CSV, a benchmark composed of charts retrieved from research papers spanning diverse scientific domains. For each chart, we manually annotate the charts with ground-truth CSVs. The task is to extract data from the charts and structure it into CSV formats, as we illustrate in Figure 2. We introduce our data collection process in Section 3.1, describe the data annotation process in Section 3.2, and formally define the CHART2CSV tasks in Section 3.3.

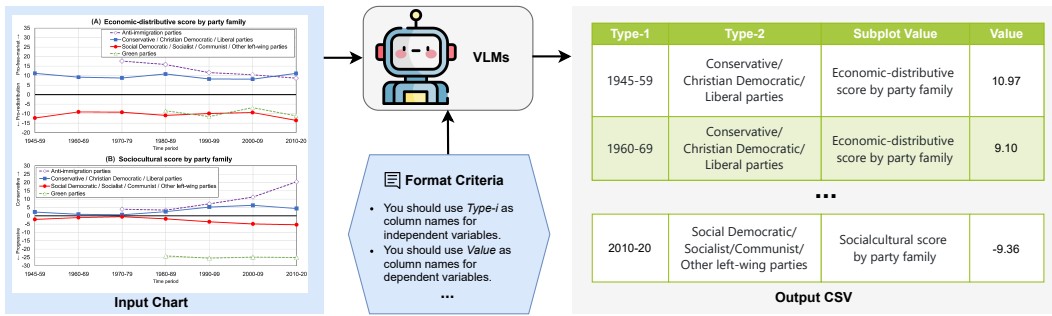

Figure 2: Example of a CHART2CSV task instance.

## 3.1 DATA COLLECTION

We collect 812 charts, 537 of which contain tables and 275 of which contain plots. For charts containing plots, we retrieve the original images embedded in the paper PDF files.

To ensure data quality, we conduct a quality control process that filters and retains only charts that are clearly identifiable by humans based on the following criteria: *(1) Resolution:* The chart must be at least $400 \times 400$ pixels in size. *(2) Layout:* There should be no overlapping data points. *(3) Data:* Each line or type in the chart must be representable with at most 50 rows. Specifically, for continuous plots, sampling 50 data points per series must be sufficient to capture the overall trends. We collect data from the following sources and domains:

**Economic and political science.** Social scientists reproduced the research findings of 110 economic and political science papers (Brodeur et al., 2024), from which we collect 532 table charts and 262 plot charts to support reproducibility analysis in these domains.

**Psychology.** The Open Science Academy reproduced 100 psychological papers (Collaboration, 2012; 2015). From these, we collect 1 table chart and 3 plot charts to capture critical data representations in psychology.

**Finance.** To investigate the capability of VLMs in extracting and understanding data visualizations in finance, we collect 1 table chart and 4 plot charts from the 27 papers published in *The Journal of Finance* (Association, 2024), a premier academic organization devoted to the study and promotion of knowledge about finance, published from 08/01/2024 to 10/31/2024.

**Biology.** To evaluate VLMs' data extraction performance in biological contexts, we collect 1 table chart and 4 plot charts from the 18 science reports in computational biology and bioinformatics published in Nature (Nature, 2024), a world's leading multidisciplinary science journal, published from 10/30/2024 to 11/1/2024

**Engineering.** To examine VLMs' understanding of engineering data, we collect 3 table charts and 2 plot charts from papers accepted to CVPR 2024 (IEEE/CVF, 2024), a leading conference in CV.

Our collection mechanism ensures that CHART2CSV represents the complexity level of charts used to visualize data in research findings. We provide detailed statistics in Table 2.

## 3.2 DATA ANNOTATION

Annotating ground-truth CSVs for CHART2CSV instances involves two phases: (1) data extraction and (2) data structuring.

To streamline data extraction, we develop a script that extracts preliminary values. For tables, it uses OCR (Singer-Vine, 2024); for plots, it detects components with OpenCV Canny (OpenCV, 2024a) and WebPlotDigitizer (WebPlotDigitizer, 2024), measures lengths using OpenCV Contours

Table 2: Statistics of the charts in CHART2CSV. For formatting reasons, tables may appear rotated 90 degrees to meet page layout constraints (Panel 6).

| Panel 1: Plot Information Density (Avg.) | | | Panel 4: Plot Style Distribution | | | |
|---|---|---|---|---|---|---|
| # Subplots | # Curves | # Data Points | Dot | Histogram | Continuous | Mixed |
| 2.03 | 3.84 | 271.33 | 164 | 64 | 36 | 11 |
| **Panel 2: Table Information Density (Avg.)** | | | **Panel 5: Plot Format Distribution** | | | |
| # Panels | # Rows | # Columns | Pixel | | Vector | |
| 1.06 | 10.67 | 10.66 | 88 | | 187 | |
| **Panel 3: Plot Extension Distribution** | | | **Panel 6: Table Orientation Distribution** | | | |
| .png | .jpg/.jpeg | .webp | Regular | | Rotated | |
| 251 | 21 | 3 | 504 | | 33 | |

(OpenCV, 2024b), and infers values w.r.t. axis tick positions. Expert annotators then review the extracted data to correct errors and fill in missing values.

In the data structuring phase, expert annotators structure the reviewed data into CSVs. To ensure consistent evaluation, we establish formatting criteria for the manual structuring process and integrate the same criteria into the task description for VLMs. The foundational formatting criteria are as follows: *(1)* Use `Type-i` as column names for independent variables, where `i` starts from 1. *(2)* Use `Value` as column names for dependent variables. *(3)* Use `-` to connect hierarchical structures, e.g., `Subplot - Subsubplot`. Each ground-truth CSV undergoes verification by a team of five, and disagreements are resolved through discussion until consensus is reached.

### 3.3 TASK FORMULATION

We formally define the tasks in CHART2CSV as follows: given the task description containing the formatting criteria and a chart, VLMs are tasked to extract all visible data from the chart and output a CSV file containing all the values. We provide an example in Figure 2. For clarity, we use *table* and *plot* to refer to different types of chart contents, and *CSV* to refer to the outputs of VLMs for the remainder of this paper.

## 4 EXPERIMENT SETUP

We introduce our selected VLMs for evaluation in Section 4.1, explain the prompting techniques in Section 4.2, and present evaluation metrics in Section 4.3.

### 4.1 MODELS

We select 16 VLMs, including all models with an average score over 70 across 11 academic benchmarks[1] and their light versions (AI2, 2024a): GPT-4o (OpenAI, 2024a), GPT-4o-mini (OpenAI, 2024b), Claude 3.5 Sonnet (Anthropic, 2024), Gemini 1.5 Pro (Google, 2024b), Gemini 1.5 Flash (Google, 2024a), Molmo 72B (AI2, 2024a), Molmo 7B-D (AI2, 2024b), Molmo 7B-O (AI2, 2024c), Qwen VL2 72B (Qwen, 2024a), Qwen VL2 7B (Qwen, 2024b), Intern VL2 LLAMA 76B (OpenGVLab, 2024b), Intern VL2 1B (OpenGVLab, 2024a), LLAVA OneVision 72B (Li et al., 2024), and LLAVA OneVision 7B (Li et al., 2024). We additionally include the recently released Claude Sonnet 4 and Claude Opus 4 (Anthropic, 2025) to obtain up-to-date insights into the capabilities of state-of-the-art VLMs.

---

[1]AI2D test (Kembhavi et al., 2016), ChartQA test (Masry et al., 2022), VQA v2.0 test (VQA, 2017), DocQA test (Mathew et al., 2021b), InfographicVQA test (Mathew et al., 2021a), TextVQA val (Singh et al., 2019), RealWorldQA (XAI, 2024), MMMU val (Yue et al., 2024), MathVista testmini (Lu et al., 2024), CountBenchQA (Beyer et al., 2024), and Flickr Count (Young et al., 2014).

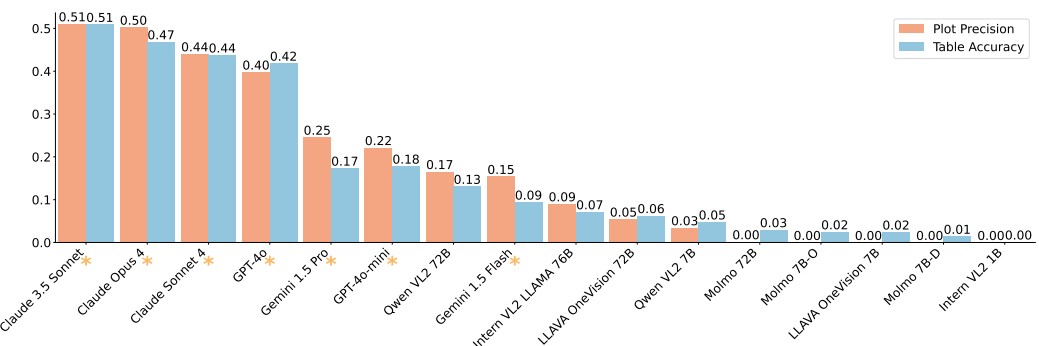

Figure 3: The overall performance of the VLMs. * indicates closed-source models.

## 4.2 PROMPTING TECHNIQUES

We use three prompting techniques: a baseline prompt, consisting only of the task description and annotation criteria; chain-of-thought (CoT) reasoning (Wei et al., 2023); and few-shot learning (Brown et al., 2020), two widely adopted techniques for VLMs. Unless otherwise specified, results are reported using the baseline prompt.

## 4.3 METRICS

We approach data extraction from tables as a classification problem, where we measure the accuracy of the extracted values, and data extraction from plots as a tracking problem, where we measure the precision of the extracted traces. We introduce our metrics in detail as follows, using $y_{pred}$ to denote the extracted values by VLMs and $y_{gt}$ to denote the ground truths.

**Tables.** We apply accuracy as the primary metric for tables. For each cell in a table, if the cell contains numeric values, it is considered accurate if $y_{\text{pred}}$ is equal to $y_{\text{gt}}$. For non-numeric values, we calculate the string matching score $s$ using fuzz.ratio (seatgeek, 2024). The cell is considered accurate if $s \geq$ threshold. The string matching score $s$ is calculated as follows:

$$s = \frac{|y_{\text{pred}}| + |y_{\text{gt}}| - lev(y_{\text{pred}}, y_{\text{gt}})}{|y_{\text{pred}}| + |y_{\text{gt}}|} \times 100,$$

where $lev(\cdot, \cdot)$ calculates the Levenshtein distance of two sequences, with $lev(y_{\text{pred}}, y_{\text{gt}}) = \frac{|y_{\text{pred}}| + |y_{\text{gt}}|}{2}$ representing moderate similarity, allowing for minor discrepancies such as small spelling errors or slight formatting differences, while still requiring a reasonable degree of similarity (Levenshtein, 1965). Thus, we set the threshold at 50. The overall table accuracy is calculated as the average accuracy of all cells in the table. If the models fail to generate valid CSVs, the accuracy of the table is considered 0.

**Plots.** We apply precision as the primary metric for plots. For continuous plots, we first discretize them by uniformly re-sampling 50 points from the models' extracted traces as $y_{pred}$. We then pair $y_{gt}$ with $y_{pred}$. For each paired value, we calculate the bounded mean absolute scaled error (MASE) $\epsilon$ as follows:

$$\epsilon = \min \left( \frac{|y_{\text{pred}} - y_{\text{gt}}|}{\max(y_{\text{gt}}) - \min(y_{\text{gt}})}, 1 \right),$$

where $\epsilon = 1$ when the absolute error between the extracted data values and ground truths exceeds the range of ground truths. For $y_{gt}$ without a paired $y_{\text{pred}}$, we set $\epsilon = 1$. The precision $p$ of the values is further calculated as $p = 1 - \epsilon$. The overall plot precision is calculated as the average precision of all data values in the plot. If the VLMs fail to generate valid CSVs, we set plot precision to 0.

## 5 EXPERIMENT RESULTS

We illustrate and analyze the overall VLM performance on CHART2CSV in Section 5.1, examine how VLMs perform on charts with different characteristics in Section 5.2, and demonstrate VLM performance using different prompting techniques in Section 5.3.

### 5.1 OVERALL VLM PERFORMANCE ON CHART2CSV

We present the table accuracy and the plot precision of all VLMs in Figure 3, from which we observe the following six major findings.

First, among all models, Claude 3.5 Sonnet performs the best, with the highest accuracy across tables at 0.51 and the highest precision across plots at 0.51. This can be attributed to the specific fine-tuning of Claude 3.5 Sonnet on complex charts. By examining outputs on continuous plots in detail, we observe that Claude 3.5 Sonnet samples data points that more accurately reflect the overall trends. For instance, in Figure 4, while GPT-4o simply uniformly samples values from the X-axis, Claude 3.5 Sonnet selects and extracts data points at each turning point, providing a better representation of the overall data distribution trends.

Second, the performance of existing VLMs remains far from satisfactory: even the best-performing model misinterprets nearly half of the data points. A common failure mode is the tendency to drop data points during extraction: as we show in Figure 5, in a plot with 57 data points, which is well below the average number of data points across the plots in CHART2CSV, even the model that extracts the most data points (Gemini 1.5 Pro) identifies an incomplete set of 55 data points, while the other models that generate valid CSVs capture only 53%–86% of the data points. We further illustrate the failure modes in structuring extracted data in detail in Appendix A.2.

Third, the trends of table accuracy and plot precision are generally consistent, with a Pearson correlation coefficient (Pearson, 1895) of 0.98. This indicates that VLMs exhibit a similar capability in extracting and structuring data from tables and plots, despite their different visual representations.

Fourth, closed-source models significantly outperform open-source models, achieving over $6\times$ higher average accuracy across tables in CHART2CSV and nearly $8\times$ higher precision in plots.

Fifth, among open-source models, Qwen VL2 72B performs the best, with the highest table accuracy at 0.13 and the highest plot precision across plots at 0.17.

Finally, within each model family, larger models perform better, showing up to $65\times$ higher accuracy for the larger model (Intern VL2 LLAMA 76B *vs.* Intern VL2 1B) across tables and up to $5\times$ higher precision across plots (Qwen VL2 72B *vs.* Qwen VL2 7B).

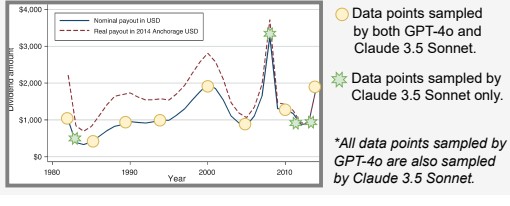 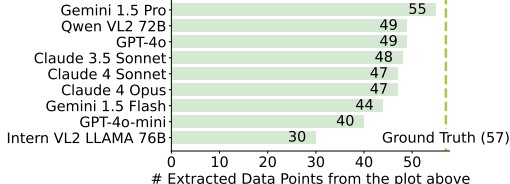

Figure 4: Sampled data points by VLMs for Figure 1 in Jones & Marinescu (2022).

Figure 5: Number of data points extracted by VLMs for Figure 1 in Laffitte & Toubal (2022).

### 5.2 VLM PERFORMANCE ACROSS DIFFERENT CHART CHARACTERISTICS

To better understand the underlying factors influencing VLM performance, we analyze how specific chart characteristics affect VLMs' ability to extract and reason over visual data. We have the following four observations.

First, as the numbers of subplots (Figure 6a) and table panels (Figure 7a) increase, the performance of the VLMs drops. This indicates that as the volume of visualized data increases, (1) the contexts processed by the VLMs become longer; (2) the outputs generated by the VLMs become more com-

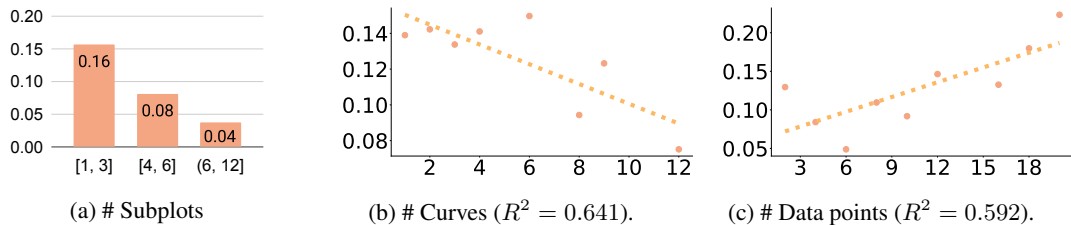

(a) # Subplots    (b) # Curves ($R^2 = 0.641$).    (c) # Data points ($R^2 = 0.592$).

Figure 6: Average plot precisions varying data densities. Features that appear in fewer than 2% of plots (fewer than 6 instances) are excluded.

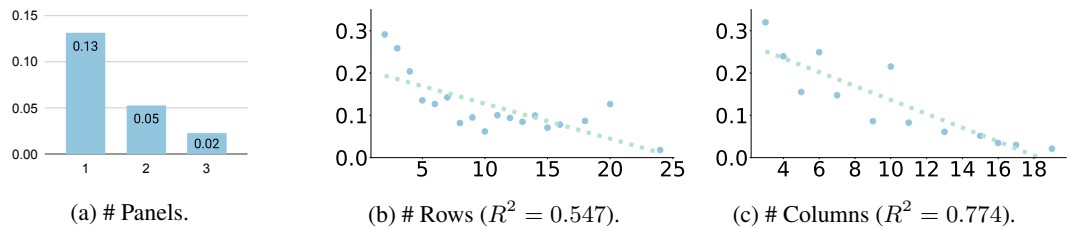

(a) # Panels.    (b) # Rows ($R^2 = 0.547$).    (c) # Columns ($R^2 = 0.774$).

Figure 7: Performance of VLMs across tables with varying data densities. Features that appear in fewer than 2% of tables (fewer than 9 instances) are excluded.

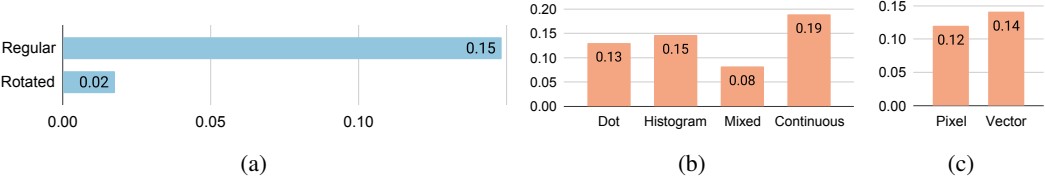

(a)    (b)    (c)

Figure 8: VLM performance on different (a) table orientations, (b) plot styles, and (c) plot formats.

plex; and (3) the additional step of first segmenting the subplots and table panels and then handling each as a separate plot or table introduces further challenges for existing VLMs.

Second, the performance of VLMs drops as the numbers of plot curves (Figure 6b), table rows (Figure 7b), and table columns (Figure 7c) increase, but enhances as the number of plot data points (Figure 6c) increases. This is because the increase in rows increases complexity at the data volume level as aforementioned, while adding columns and curves further requires interpreting the relations among different features of the data points. The observed improvement in precision with a higher number of data points in plots can be attributed to the fact that additional data points provide mutual references, enabling VLMs to extract values more accurately.

Third, we can observe from Figure 8a that the VLMs fail to perform as well on rotated tables, with accuracy dropping to nearly 1/10 as regularly displayed tables. Specifically, the accuracy on rotated tables for all open-source VLMs drops to 0%. This indicates that current VLMs struggle to recognize formatting variations in tables and do not generalize well to unconventional data visualizations in tabular forms.

Finally, we can observe from Figures 8b and 8c that the plot precision of VLMs remains consistent across different plot types and forms. This suggests that, unlike tables displayed in various formats, VLMs are less affected by variations in plot display and are capable of extracting essential data information from plots regardless of the visualization styles and forms.

### 5.3 VLM PERFORMANCE USING DIFFERENT PROMPTING TECHNIQUES

We evaluate the five best-performing VLMs from the OpenAI and Claude families on CHART2CSV under varying numbers of few-shot examples (0, 1, 2, and 3), both with and without CoT. We

Table 3: Performance of VLMs using different prompting techniques. The optimal results are highlighted. Average plot precision is 0.41, 0.42, 0.43, and 0.43 for 0-, 1-, 2-, and 3-shot settings, respectively, while average table accuracy is 0.45, 0.56, 0.58, and 0.60. The average plot precision is 0.43 without CoT and 0.41 with CoT, while the average table accuracy is 0.53 without CoT and 0.56 with CoT.

| Model | Prompting Strategy | Plot Precision | | | | Table Accuracy | | | |
|---|---|---|---|---|---|---|---|---|---|
| | | Number of Shots | | | | | | | |
| | | 0 | 1 | 2 | 3 | 0 | 1 | 2 | 3 |
| Claude 3.5 Sonnet | w/o CoT | 0.5107 | 0.5308 | 0.5315 | 0.5327 | 0.5094 | 0.6565 | 0.7082 | 0.7462 |
| | CoT | 0.4872 | 0.4984 | 0.5235 | 0.4952 | 0.5615 | 0.6980 | 0.7659 | 0.7693 |
| Claude Opus 4 | w/o CoT | 0.5028 | 0.5046 | 0.5228 | 0.5235 | 0.4687 | 0.5412 | 0.6329 | 0.6456 |
| | CoT | 0.4836 | 0.4966 | 0.4815 | 0.4976 | 0.6145 | 0.5543 | 0.6435 | 0.6240 |
| Claude Sonnet 4 | w/o CoT | 0.4401 | 0.4212 | 0.4537 | 0.4204 | 0.4379 | 0.6068 | 0.5831 | 0.6200 |
| | CoT | 0.3557 | 0.3761 | 0.4126 | 0.4247 | 0.5691 | 0.5914 | 0.6550 | 0.6694 |
| GPT-4o | w/o CoT | 0.3980 | 0.4228 | 0.4251 | 0.4296 | 0.4187 | 0.6068 | 0.6419 | 0.6496 |
| | CoT | 0.3879 | 0.4028 | 0.3906 | 0.4350 | 0.4880 | 0.6086 | 0.6203 | 0.5958 |
| GPT-4o-mini | w/o CoT | 0.2216 | 0.2622 | 0.3076 | 0.2570 | 0.1783 | 0.3692 | 0.3162 | 0.3595 |
| | CoT | 0.2676 | 0.2519 | 0.2793 | 0.2861 | 0.2594 | 0.3609 | 0.2634 | 0.2928 |

show the results in Table 3 and report the full performance of all VLMs under different prompting techniques in Appendix A.3. Across models, more sophisticated prompting techniques consistently improves performance relative to the baseline prompt, with the strongest results typically obtained using 3-shot examples. Our key findings are as follows:

**Increasing few-shot examples improves performance, but gains diminish beyond the first example.** Both plot precision and table accuracy improve as the number of few-shot examples increases. Notably, table accuracy improves more than plot precision, as the inherent structure of tables allows few-shot examples to generalize more effectively. However, the rate of improvement decreases as the number of examples grows, with the largest jump occurring between 0-shot and 1-shot. This suggests that the capacity of VLMs to exploit in-context learning is limited, even when provided with more diverse examples. Thus, advancing VLM capability in chart data extraction and structuring requires approaches beyond simply scaling the number of few-shot examples.

**CoT does not yield significant improvements and can even harm performance on plots.** We interpret this result from three perspectives: (1) *Task-level:* CHART2CSV tasks involve two stages: extracting data points from charts and organizing them as structured tables. While CoT primarily aids the latter by providing reasoning paths as templates, existing VLMs already struggle with the extraction stage, limiting overall benefits. (2) *Data-level:* For plots, CoT often reduces performance because the underlying data is less structured and the chart displays in CHART2CSV are highly diverse. As a result, CoT reasoning paths fail to generalize across tasks. (3) *Model-level:* Although CoT prompts help VLMs follow high-level reasoning steps, they struggle to adapt these steps to the specific requirements of different charts. As shown in Appendix A.3, our analysis of model traces reveals that VLMs often execute the general outline but fail to complete the fine-grained reasoning steps accurately.

## 6 CONCLUSION

In this work, we introduce CHART2CSV, a benchmark designed to evaluate the ability of VLMs to accurately and comprehensively extract data points from complex charts and convert them into structured tables. We assess 16 VLMs, comprising 7 closed-source and 9 open-source models, on CHART2CSV. Claude 3.5 Sonnet achieves the optimal performance, with the highest accuracy of 0.51 across all tables and the highest precision of 0.51 across all plots, indicating that nearly half of the data points are misinterpreted. We further evaluate a wide range of prompting techniques and find that they offer only limited improvements on CHART2CSV tasks. Based on these empirical results and analysis, we conclude that existing VLMs lack the capability to faithfully convert complex charts into structured tables, highlighting the need to build more powerful VLMs.

## 7 REPRODUCIBILITY STATEMENT

CHART2CSV is publicly available at `https://anonymous.4open.science/r/figure-to-data-2FB5/`. Evaluation code to reproduce all reported results is publicly available at `https://anonymous.4open.science/r/figure-to-data-code-C4EB/`.

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

# A APPENDIX

## A.1 VLMS OUTPERFORM TRADITIONAL CV LIBRARIES ON CHART2CSV TASKS.

We explore the limitations of using generic CV libraries and software engineering methods for tasks in CHART2CSV. We use OpenCV OpenCV (2024b;a) to extract data points from plots and identify the following limitations: (1) OpenCV fails to detect axes (Figure 9a), (2) it fails to recognize error bars (Figure 9b), (3) it misinterprets noises, such as additional lines caused by printing artifacts, in plots (Figure 9a), and (4) it cannot differentiate between gridlines and actual bars (Figure 9c).

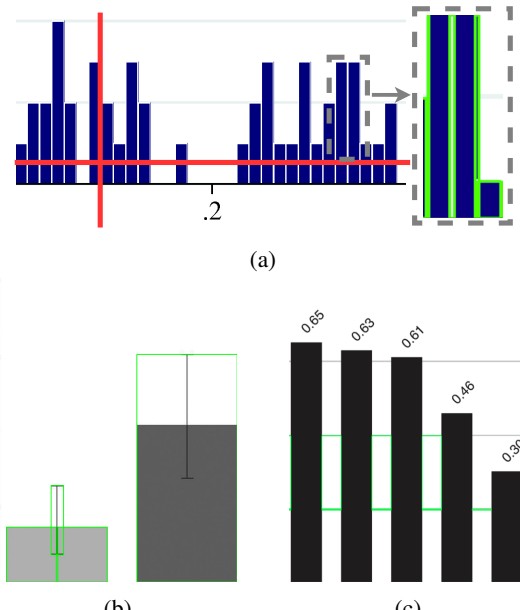

Figure 9: Examples of failure of the OpenCV library in extracting data points from plots.

To illustrate these limitations in detail, we compare the outputs of OpenCV and Claude 3.5 Sonnet on the chart in Figure 9a. First, because OpenCV cannot correctly identify the axes, the extracted values are inaccurate. For example, for Type-1 with $y_{gt} = 0.0099$ and Claude 3.5 Sonnet accurately extracts $y_{pred} = 0.01$, OpenCV erroneously generates $y_{pred} = 0.035$ by matching the wrong bar. Second, OpenCV extracts additional data points due to extra lines in the chart caused by printing artifacts, which it mistakenly interprets as bars. In this case, OpenCV falsely extracts 5 consecutive data points with identical values, while the CSV file generated by Claude 3.5 Sonnet contains the correct number of data points and matches the bars. Thus, compared to generic CV libraries like OpenCV, VLMs such as Claude 3.5 Sonnet demonstrate greater accuracy in detecting axes, gridlines, and error bars, as well as filtering out minor noise in plots.

## A.2 VLMS STRUGGLE TO STRUCTURE DATA WITH COMPLEX RELATIONAL INFORMATION

Following our analysis in Section 5.1 on the limitations of data extraction, we turn to data structuring, showing that it remains a major challenge even in charts with tables whose data are clearly extracted. Unlike standard tables like dataframes, where relationships exist across columns, the tables presented in research findings may include additional rows with supplementary information, such as confidence intervals (Figure 10). We observe that the performance of different VLMs varies significantly in handling such cases. For instance, when extracting data points from the table in Figure 10, all 3 models (GPT-4o, Claude 3.5 Sonnet, and Qwen VL2 72B) that successfully generate valid CSVs correctly parse 4 columns. However, only GPT-4o fills in all values accurately, achieving a table accuracy of 100%. Both Claude 3.5 Sonnet and Qwen VL2 72B fail to retrieve values from the Parameter column and only partially capture strings in the Prior column. Specifically, Claude 3.5 Sonnet achieves an accuracy of 11%, the lowest among all of its generated valid tables, while Qwen VL2 72B achieves an accuracy of 42%.

## A.3 COMPREHENSIVE EVALUATION OF ADVANCED PROMPTING TECHNIQUES

We examine the effects of 2-shot CoT, an effective setting on existing chart benchmarks Lu et al. (2024), as well as 0-shot CoT as a reference. We evaluate the effectiveness of 0-shot CoT across all 16 VLMs and 2-shot CoT across 11 VLMs, excluding those that do not support images as part of in-context learning (i.e., the Molmo and Llava families). We present the full results in Table 4 and summarize our key observations as follows:

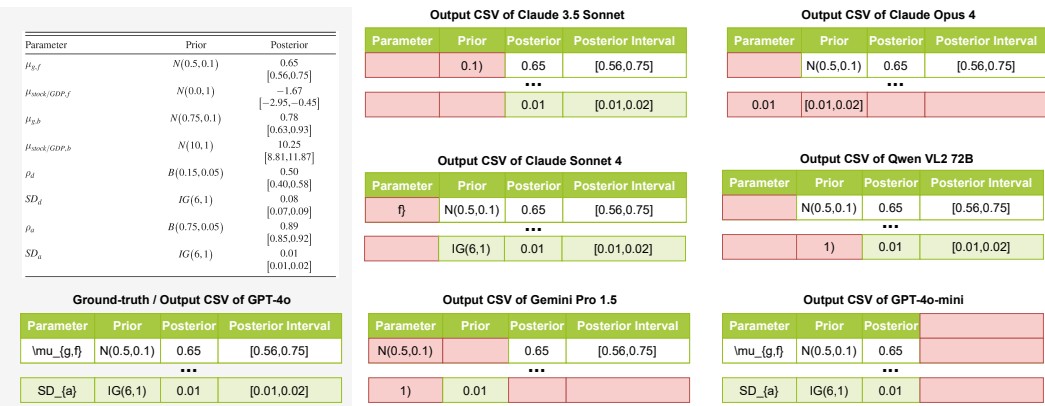

Figure 10: Output CSVs of different VLMs for Table 2 in Guerron-Quintana et al. (2023).

Table 4: Overall performance of the VLMs using various prompting techniques. * indicates closed-source models. The best performance for each VLM is highlighted. Average plot precision is 0.18 (baseline), 0.17 (0-shot CoT), and 0.25 (2-shot CoT); average table accuracy is 0.17 (baseline), 0.21 (0-shot CoT), and 0.34 (2-shot CoT).

| Model | Plot Precision | | | Table Accuracy | | |
|---|---|---|---|---|---|---|
| | Baseline | 0-shot CoT | 2-shot CoT | Baseline | 0-shot CoT | 2-shot CoT |
| Claude 3.5 Sonnet * | 0.5107 | 0.4872 | 0.5235 | 0.5094 | 0.5615 | 0.7659 |
| Claude Opus 4 * | 0.5028 | 0.4836 | 0.4815 | 0.4687 | 0.6145 | 0.6435 |
| Claude Sonnet 4 * | 0.4401 | 0.3557 | 0.4126 | 0.4379 | 0.5691 | 0.6550 |
| GPT-4o * | 0.3980 | 0.3879 | 0.3906 | 0.4187 | 0.4880 | 0.6203 |
| Gemini 1.5 Pro * | 0.2458 | 0.3264 | 0.2052 | 0.1736 | 0.3039 | 0.3680 |
| GPT-4o-mini * | 0.2216 | 0.2676 | 0.2793 | 0.1783 | 0.2594 | 0.2634 |
| Qwen VL2 72B | 0.1656 | 0.1657 | 0.3018 | 0.1311 | 0.0968 | 0.0871 |
| Gemini 1.5 Flash * | 0.1541 | 0.1813 | 0.0877 | 0.0943 | 0.2230 | 0.2263 |
| Intern VL2 LLAMA 76B | 0.0903 | 0.0203 | 0.0217 | 0.0716 | 0.0764 | 0.0816 |
| LLAVA OneVision 72B | 0.0546 | 0.0644 | / | 0.0627 | 0.0617 | / |
| Qwen VL2 7B | 0.0344 | 0 | 0 | 0.0480 | 0.0369 | 0.0146 |
| Molmo 72B | 0 | 0 | / | 0.0017 | 0 | / |
| Molmo 7B-O | 0 | 0 | 0 | 0.0249 | 0 | / |
| LLAVA OneVision 7B | 0 | 0 | 0 | 0.0243 | 0.0720 | / |
| Molmo 7B-D | 0 | 0 | 0 | 0.0144 | 0.0089 | / |
| Intern VL2 1B | 0 | 0 | 0 | 0.0011 | 0.0002 | 0.0002 |

**Advanced prompting techniques improves VLMs' performance on CHART2CSV, but not to a satisfactory level.** In general, for both plots and tables, VLMs prompted with 0-shot CoT outperform those using the baseline prompt, and VLMs prompted with 2-shot CoT significantly outperform those using 0-shot CoT and the baseline prompt. Specifically, two-shot CoT increases Claude 3.5 Sonnet's plot precision to 0.5235 and table accuracy to 0.7659, with the latter representing a relative improvement of over 50%. Despite these gains, the absolute performance remains limited. Claude 3.5 Sonnet is the only model to surpass 70% accuracy on tables, and its plot interpretation, although the best across all models and prompting settings, still misidentifies nearly half of the data points. These results suggest that the low performance of VLMs reflects systematic deficiencies in the models themselves, which cannot be fully addressed through prompting alone. Model fine-tuning or architectural enhancements is required to improve VLMs' performance on CHART2CSV tasks.

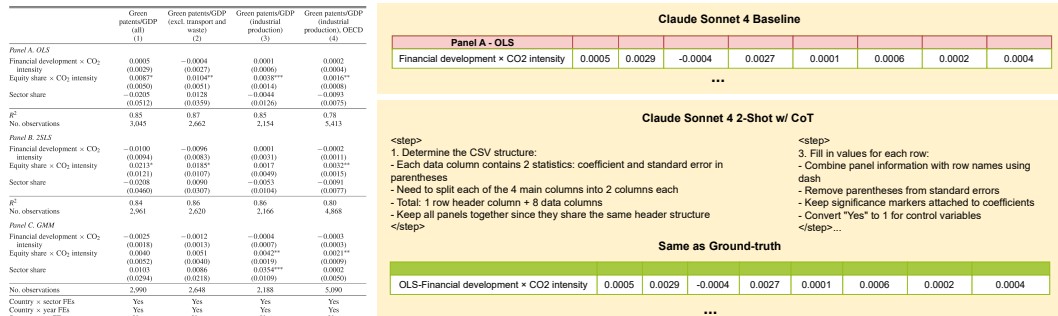

(a) Outputs of Claude Sonnet 4 with baseline prompt and 2-shot CoT prompt for Table 6 of Haas & Popov (2023).

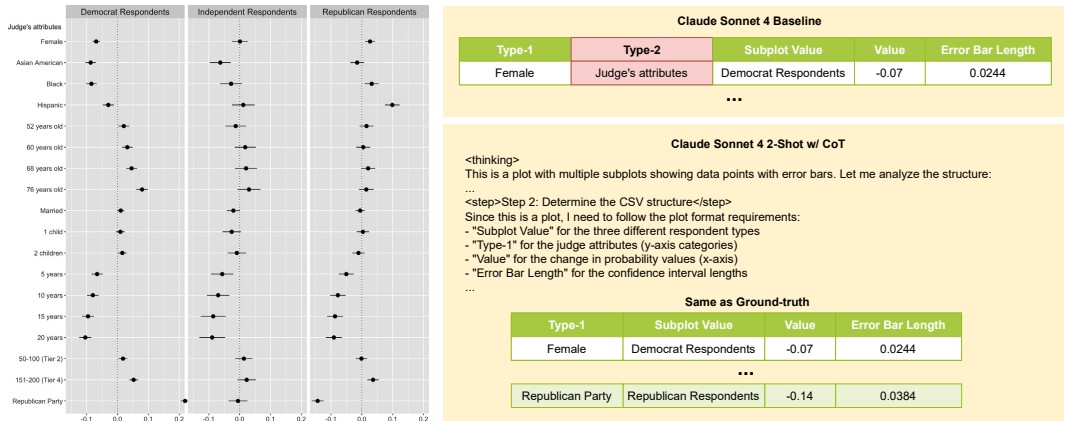

(b) Outputs of Claude Sonnet 4 with baseline prompt and 2-shot CoT prompt for Figure 2 of Ono & Zilis (2022).

Figure 11: Comparison of VLM performance with baseline and advanced prompting techniques.

**Advanced prompting techniques do not shift relative performance.** From Table 4, we observe that the ranking of the best-performing VLMs for plot precision remains consistent with the ranking when using the baseline prompt. For table accuracy, the performance order also largely aligns with that of the baseline prompt, except for two cases: Gemini 1.5 Flash outperforms Qwen VL2 72B when prompted with 2-shot CoT, and the performance of Molmo 72B drops significantly due to its instability in generating valid CSVs when following the reasoning path of CoT prompts.

**Larger and more capable VLMs benefit most from complex prompting.** Table 4 shows that 2-shot CoT provides the most substantial performance gains for closed-source models and large open-source models. These advanced models are better equipped to handle longer inputs and complex reasoning. For example, Qwen VL2 72B nearly doubles its plot precision compared to the baseline prompt, while Claude 3.5 Sonnet improves its table accuracy by 0.2 in absolute terms. In contrast, smaller models show little to no improvement and can even perform worse under CoT prompting, reflecting their limitations in processing and understanding more elaborate input structures.

**Prompting techniques yield stronger gains on tables than on plots.** The performance improvements achieved through various prompting techniques are more evident and consistent for tables than plots. Specifically, the average table accuracy across all models doubles when using 2-shot CoT compared to the baseline prompt. Moreover, for the majority of advanced VLMs, the highest table accuracy is achieved under the most complex prompting configuration. This implies that the reasoning paths are particularly beneficial for tables because *(1)* Table data is easier to identify, whereas extracting data from plots requires VLMs to accurately interpret visual values, adding an additional level of complexity. The reasoning paths designed to help VLMs determine output structures and formats do not effectively address this issue, highlighting the potential of CHART2CSV

data to improve VLM capabilities as a promising direction for future work. *(2)* Tables are typically more well-structured, allowing VLMs to generalize more effectively from the provided few-shot examples. In contrast, plots are less structured, and few-shot examples can sometimes be misleading. For example, as seen in Table 4, Gemini 1.5 Pro and Gemini Flash achieve their best performance when prompted with 0-shot CoT, but their performance drops with 2-shot CoT.

**Advanced prompting techniques help VLMs capture inter-column semantics.** We analyze cases where plot precision and table accuracy increase significantly when switching from the baseline prompt to more complex prompting strategies. Our findings suggest that these improvements are largely due to the model's enhanced ability to recognize structured relationships among columns, enabling more accurate formulation of the output schema. As illustrated in Figure 11, the 2-shot CoT prompt helps the model capture critical structural information. In Figure 11a, the model correctly encodes panel-level distinctions that were missed under the baseline prompt. Similarly, in Figure 11b, the model with 2-shot CoT correctly identifies a shared value as a column entry, which the baseline prompt fails to do.

### A.4  USE OF LARGE LANGUAGE MODELS (LLMS)

No LLMs were used in the ideation, writing, or preparation of this paper. All content was conceived, drafted, and revised solely by the authors.

