# OpenReview forum: "Chart2CSV: Can VLMs Faithfully Convert Complex Charts into Structured Tables?"
_ICLR.cc/2026/Conference — ICLR 2026 Conference Withdrawn Submission_

### Official Review · Reviewer_7GR9 · 2025-10-25

**Soundness:** 2
**Presentation:** 3
**Contribution:** 2
**Rating:** 6
**Confidence:** 3

**Summary:**

This paper introduces **CHART2CSV**, a new benchmark designed to evaluate the ability of Vision-Language Models (VLMs) to faithfully convert complex charts into structured CSV tables. The authors argue that existing benchmarks are insufficient, as they use overly simplified charts and lack comprehensive metrics for both completeness and accuracy. The benchmark itself, which is the primary contribution, consists of **812 charts** (275 plots, 537 tables) sourced from real-world research papers across five scientific domains. A comprehensive evaluation of 16 VLMs on CHART2CSV revealed a significant deficiency in current models; The analysis further highlights that model performance drops significantly with increased chart complexity, especially on unconventional formats like **rotated tables**, and that advanced prompting techniques like CoT offer only limited improvements, suggesting core model deficiencies.

**Strengths:**

This paper's core strength is its contribution of the **CHART2CSV benchmark**, a high-quality and significant dataset that is exceptionally useful. Sourced from 812 complex, real-world scientific papers, it moves beyond the oversimplified charts in prior work. The benchmark's main significance lies in its clarity in revealing the profound limitations of current state-of-the-art VLMs, which, as the paper shows, misinterpret nearly half the data points. Furthermore, the work provides a robust analysis of *why* these models fail, identifying specific, actionable weaknesses such as handling increased chart complexity and unconventional formats like rotated tables. In doing so, this paper provides the community with both a crucial diagnostic tool and a clear direction for future research aimed at improving these fundamental model capabilities.

**Weaknesses:**

A significant limitation of this study is the scope of the evaluated models. The paper focuses exclusively on general-purpose VLMs (e.g., Claude 3.5 Sonnet, GPT-4o) and concludes that *all* VLMs are deficient in this area. However, this conclusion feels premature as the evaluation completely omits a critical category of specialized models designed explicitly for chart and document understanding. For instance, the comparison lacks models like **ChartLlama** (Han et al., 2023), a multimodal LLM fine-tuned for chart-specific and pure CV models (Junyu et al., 2021). Furthermore, since the task is heavily reliant on robust text and data extraction from complex layouts, the benchmark should have been tested against state-of-the-art OCR-focused models, such as **DeepSeek-OCR** (Wei et al., 2025) or the unified end-to-end model from Wei et al. (2024). Without including these specialist models, it is unclear whether the observed failures are a fundamental limitation of all current VLM architectures or simply a failure of general-purpose models that more specialized approaches may have already solved. The paper's claims would be much stronger if it included this crucial baseline comparison.

Reference:

Wei H, Sun Y, Li Y. DeepSeek-OCR: Contexts Optical Compression[J]. arXiv preprint arXiv:2510.18234, 2025. H. Wei, C. Liu, J. Chen, J.

Wang, L. Kong, Y. Xu, Z. Ge, L. Zhao, J. Sun, Y. Peng, et al. General ocr theory: Towards ocr-2.0 via a unified end-to-end model. arXiv preprint arXiv:2409.01704, 2024.

Han Y, Zhang C, Chen X, et al. Chartllama: A multimodal llm for chart understanding and generation[J]. arXiv preprint arXiv:2311.16483, 2023.

Luo J, Li Z, Wang J, et al. Chartocr: Data extraction from charts images via a deep hybrid framework[C]//Proceedings of the IEEE/CVF winter conference on applications of computer vision. 2021: 1917-1925.

**Questions:**

My primary questions concern the scope and evaluation of the benchmark.

First, could the authors clarify why specialized chart and document models (e.g., ChartLlama, DeepSeek-OCR) were omitted from the evaluation? This exclusion makes it difficult to ascertain if the observed poor performance is a fundamental limitation of all VLMs or just a failure of the general-purpose models tested.

Finally, how do the current metrics differentiate between structural errors (e.g., wrong columns) and value errors?

---

> ### Author Response · Authors · 2025-11-27
> **Response to Reviewer 7GR9**
>
> Dear Reviewer 7GR9,
>
> Thank you for your detailed summary and constructive feedback. We are glad that you find our work valuable in contributing a high-quality real-world benchmark, impactful in revealing fundamental VLM limitations, and informative in providing actionable analysis for future research. Below, we address the mentioned weaknesses:
>
> **W1, Q1:** We agree that specialized chart models are important reference points. However, models such as ChartLlama and DeepSeek-OCR are primarily optimized for point-level extraction rather than structural chart reconstruction, which is the central focus of our benchmark. Our task requires models not only to extract values but also to infer and represent the structural relationships between them (column semantics, multi-series grouping, hierarchical axes, panel layouts, etc.).
>
> To clarify this point, we evaluated DeepSeek-OCR on Chart2CSV. The results show that while DeepSeek-OCR can identify some textual elements, it fails to produce coherent structured outputs:
>
> - Plot precision: 0.0
>
> - Table accuracy: 0.0
>
> - Valid CSV outputs: 1.1%
>
> These results suggest that specialized OCR-oriented systems do not transfer well to our setting because they lack the structural reasoning required to reconstruct full charts.
>
> **Q2:** Our benchmark uses a unified cell-wise matching metric, which means that we do not explicitly separate structural errors from value errors. Instead, both types of mistakes naturally manifest as cell mismatches.
>
> Regards,
>
> Paper 14065 Authors

---

> > ### Comment · Reviewer_7GR9 · 2025-11-28
> >
> > Thanks for your response — I’ll take your feedback into consideration when rating.

---

### Official Review · Reviewer_MkjL · 2025-10-30

**Soundness:** 3
**Presentation:** 2
**Contribution:** 2
**Rating:** 2
**Confidence:** 4

**Summary:**

This paper introduces CHART2CSV, a new benchmark designed to evaluate Vision-Language Models' (VLMs) capability to accurately extract all data points from research paper charts and convert them into structured CSV tables. The benchmark addresses the critical limitations of existing evaluations by including complex, real-world charts and requiring advanced VLM capabilities like perception, reasoning, planning, and long-form output generation. Empirical results demonstrate that state-of-the-art VLMs perform poorly on CHART2CSV, with the best model achieving only 51% accuracy, highlighting significant limitations in current models for reliable data digitization from visualizations.

**Strengths:**

1. The motivation for this study is clear. Although advanced visual language models (VLMs) demonstrate strong performance in general vision tasks, they continue to encounter considerable challenges in accurately extracting complete datasets from charts and organizing them into well-structured tables.
2.  The paper is easy to read and understand.

**Weaknesses:**

It is essential to introduce a new benchmark that better aligns with real-world applications. However, the Chart2CSV dataset falls short in several key areas.

1. With fewer than 1,000 charts, it lacks the diversity and complexity required for robust evaluation. In contrast, datasets like VG-DCU feature a wide range of chart categories, including Area, Counter, Violin, and Sankey charts.

2. Additionally, there are concerns regarding the accuracy of annotations. The reliance on manual annotation from academic papers further limits the scalability of the benchmark. Platforms such as Plotly allow for the retrieval of charts in various formats (e.g., PNG, SVG) along with raw CSV data. SVG-format charts inherently contain rich structural information, which can be leveraged to generate accurate labels using automated scripts, eliminating the need for manual annotation.

3. Lastly, while the current visual language model (VLM) has been evaluated on a small benchmark, its contribution remains insufficient to meet the standards required for top-tier conferences.

**Questions:**

This work presents two key areas for improvement:

1. The benchmark requires further refinement, especially in terms of its size and diversity.

2. The authors are encouraged to propose methods for enhancing the current visual language model (VLM) on the Chart2CSV task

---

> ### Author Response · Authors · 2025-11-27
> **Response to Reviewer MkjL**
>
> Dear Reviewer MkjL,
>
> Thank you for your detailed summary and constructive feedback. We are glad that you find our work well-motivated, clearly presented, and addressing an important challenge in chart data extraction. Below, we address the mentioned weaknesses:
>
> **W1, Q1:** Our benchmark focuses on full-fidelity chart reconstruction, which differs fundamentally from chart digitalization benchmarks like VG-DCU, as we compare in Table 1 and Section 2. Chart digitalization requires annotating only a small number of representative points, whereas Chart2CSV requires extracting all data points, determining their relationships, and organizing them into a fully structured table. Full chart reconstruction necessarily requires expert manual annotation, and scaling such a benchmark is inherently constrained by human labor.
>
> **W2:** The reviewer’s claim that SVG formats can yield accurate labels “eliminating manual annotation” does not hold for our task. Real-world scientific figures are complex and far from programmatically perfect. While SVG extraction can recover shapes and coordinates, to name just a few limitations, it cannot infer underlying data values, normalize heterogeneous scales, interpret chart semantics, or group points into the correct tabular structure. Therefore, synthetic pipelines or SVG-based scripts cannot serve as substitutes for our problem setting.
>
> **W3, Q2:** As a benchmark paper, our goal is to diagnose model limitations, not to propose new training or architectural strategies. Nevertheless, we have already explored a wide range of prompting techniques in Section 5.3, but even with these methods, existing VLMs fail to faithfully reconstruct full charts. This highlights the need for future methods specifically designed for structured chart reconstruction.
>
> Regards,
>
> Paper 14065 Authors

---

### Official Review · Reviewer_SFjA · 2025-10-31

**Soundness:** 2
**Presentation:** 3
**Contribution:** 2
**Rating:** 2
**Confidence:** 4

**Summary:**

This paper introduces CHART2CSV, a benchmark designed to evaluate vision-language models' (VLMs) ability to extract data from complex charts in research papers and convert them into structured CSV tables. The benchmark comprises 812 charts (537 tables, 275 plots) from 5 scientific domains, with expert-validated ground truth CSVs. The authors evaluate 16 VLMs using various prompting techniques and find that even the best-performing model (Claude 3.5 Sonnet) achieves only 0.51 accuracy/precision, misinterpreting nearly half of the data points. The paper argues that existing VLMs are insufficient for automating chart-to-table conversion tasks.

**Strengths:**

1. **Rigorous annotation process and practical importance**: The ground truth annotation involves preliminary automated extraction followed by expert review and consensus among five annotators, ensuring high-quality labels. The problem of extracting data from research charts where original data is unavailable has clear practical applications in scientific reproducibility and accessibility.

2. **Comprehensive evaluation with insightful analysis**: The paper evaluates 16 VLMs across multiple prompting techniques and provides systematic analysis of performance characteristics (e.g., scaling effects, chart complexity factors, rotated tables). The findings reveal critical limitations, such as catastrophic failure on rotated tables (0% accuracy for open-source models).

3. **Clear presentation with concrete examples**: The paper is well-written with effective use of examples (Figure 1) to illustrate model failure modes. The task formulation is clear and the experimental setup is transparently described.

**Weaknesses:**

1. **Insufficient differentiation from CharXiv and missing evaluation of reasoning models**: The paper cites CharXiv (NeurIPS 2024) but fails to explain key differences despite both focusing on research paper charts. Table 1 marks CharXiv with "X" for "Verified (Exhaustive)" without clarification. More critically, CharXiv's online leaderboard shows reasoning models (including OpenAI o3) perform significantly better on chart understanding tasks, yet this paper completely omits reasoning-capable models (o1-series released September 2024, Gemini 2.5 Pro released June 2024) that were available before the deadline. This omission undermines the paper's central claim about VLM limitations and represents a major gap in the evaluation.

2. **Weak justification for real-world charts and unsubstantiated claims**: The paper emphasizes using charts from "real research findings" but never articulates what specific complexity cannot be replicated with synthetic data generation (e.g., programmatic variations using matplotlib, pgfplots). The claim of "real-world complexity" remains vague. Additionally, the paper makes unsubstantiated claims like "This can be attributed to the specific fine-tuning of Claude 3.5 Sonnet on complex charts" (line 335) without evidence. Anthropic has not confirmed chart-specific fine-tuning.

3. **Methodological issues and data imbalance**: The metric terminology is confusing. "Precision" for plots actually measures both accuracy and completeness since missing points receive penalties. No inter-annotator agreement metrics are provided despite claims of consensus-based annotation. The dataset is heavily skewed toward economic/political science (794/812 charts), with only 18 charts from psychology, finance, biology, and engineering combined, undermining cross-domain generalization claims.

**Questions:**

1. **CharXiv and reasoning models**: Can you provide detailed comparison with CharXiv's verification process and explain what makes yours more "exhaustive"? Why were reasoning-capable models (o1, Gemini 2.5 Pro) excluded despite being available before submission? Given CharXiv's results showing strong performance from reasoning models, how might your conclusions change?

2. **Real-world vs. synthetic justification**: What specific properties of real-world research charts cannot be replicated through synthetic generation with programmatic style variations? Can you provide concrete examples of complexity that synthetic approaches would fundamentally miss?

3. **Methodological details**: What were inter-annotator agreement scores before consensus? Why use "precision" terminology when the metric includes recall-like penalties? Given the domain imbalance (97.8% economic/political science), how confident are you in cross-domain generalization?

---

> ### Author Response · Authors · 2025-11-27
> **Response to Reviewer SFjA**
>
> Dear Reviewer SFjA,
>
> Thank you for your detailed summary and constructive feedback. We are glad that you find our work rigorous in its annotation process, practically meaningful for real-world scientific reproducibility, comprehensive in its evaluation and analysis, and clearly presented with concrete examples. Below, we address the mentioned weaknesses:
>
> **W1, Q1:** CharXiv and our benchmark target fundamentally different problem settings. CharXiv evaluates reasoning over chart content, where the model answers natural-language questions given a chart image. In contrast, our task requires full structured reconstruction of the chart (all datapoints, axes, series, metadata) and evaluates the fidelity of the reconstruction using point-level precision metrics.
>
> To address the concern about model coverage, we have now added two of the strongest publicly available models, GPT-5 and o4-mini. Their performance is shown below. These results further reinforce our key finding: even top-tier VLMs struggle with faithfully extracting structured data from scientific charts. Stronger reasoning capabilities alone do not yield better reconstruction precision since accurate data-point extraction remains a critical bottleneck.
>
> | Model            | Plot Precision | Table Accuracy |
> |------------------|----------------|----------------|
> | GPT-5            | 0.42          | 0.36          |
> | o4-mini          | 0.44        | 0.49          |
>
> **W2, Q2:** We have removed the speculative statement regarding Claude 3.5 Sonnet.
>
> Regarding real-world charts, we clarify that our motivation is not merely stylistic diversity but structural heterogeneity. For example, multi-layered statistical annotations, domain-specific panel decompositions, irregular axis transformations, mixed categorical encodings, etc, are frequently found in published papers. These elements are difficult to synthesize with conventional script-based generators such as matplotlib or pgfplots, which tend to produce idealized, regularized figures.
>
> **W3, Q3:** Our “precision” metric is a strict one that penalizes both incorrect extractions and missing points; we will rename it to *point-wise reconstruction precision* to avoid confusion with classical precision/recall terminology.
>
> As we report in the paper lines 239-240, all annotations were performed by a team of five trained annotators, iterating until full agreement was reached.
>
> Finally, we acknowledge that the dataset is currently skewed toward economics/political science. Our intention is not to claim broad cross-domain generalization but to evaluate VLMs on a class of charts that appear frequently in empirical social-science literatures.
>
> Regards,
>
> Paper 14065 Authors

---

### Official Review · Reviewer_Cs4B · 2025-11-03

**Soundness:** 3
**Presentation:** 3
**Contribution:** 3
**Rating:** 6
**Confidence:** 3

**Summary:**

The paper introduces the Chart2CSV benchmark that evaluates how effectively vision-language models (VLMs) can extract and structure data from complex research charts into CSV format. The authors argue that existing benchmarks are insufficient because they use oversimplified charts that fail to comprehensively assess critical VLM capabilities like perception, reasoning, and long-form output generation. Chart2CSV comprises 812 charts from five scientific domains, each paired with an expert-validated ground-truth CSV, and evaluation reveals that even the top-performing VLM, Claude 3.5 Sonnet, misinterprets nearly half the data, underscoring significant deficiencies in current VLM capabilities.

**Strengths:**

1. The task of extracting structured data from chart images is a well-motivated and important problem.

2. Created a benchmark from real-world data. With the recent trend of releasing so many synthetic benchmarks, this is a decent contribution.

3. The annotation process is of high quality: uses CV tools for a first pass, followed by expert manual validation.

4. Conducted extensive experiments to evaluate the capabilities and limitations of various LLMs.

5. The dataset is also released. This helps ensure reproducibility.

**Weaknesses:**

1. Lack of a Human Baseline: The paper shows that VLMs are not good at this task (around 50% accuracy). However, a human baseline would be invaluable in such complex benchmarks.

2. Inter Annotator Agreement is not reported. This creates concern on the annotation process.

3. Lack of in-depth discussion on what plot types are added, and what are missing.

4. Missing strong models like Gemini-2.5 (pro and flash), GPT-5, etc.

**Questions:**

Address the mentioned weaknesses.

---

> ### Author Response · Authors · 2025-11-27
> **Response to Reviewer Cs4B**
>
> Dear Reviewer Cs4B,
>
> Thank you for your detailed summary and constructive feedback. We are glad that you find our work well-motivated, based on high-quality real-world annotations, supported by extensive model evaluations, and beneficial due to the public release of our dataset. Below, we address the mentioned weaknesses:
>
> **W1:** The charts we select are from papers that have been reproduced by experts [1], meaning that human experts can extract the correct underlying data from these charts, establishing an upper bound for achievable performance.
>
> **W2:** As we report in the paper lines 239-240, all annotations were performed by a team of five trained annotators, iterating until full agreement was reached.
>
> **W3:** In Table 2, we outline a wide range of diversity in plot types, including information density (panel 1), extension (panel 3), style (panel 4), and format (panel 5).
>
> **W4:** We added evaluations of GPT-5 and o4-mini, two of the strongest publicly available models, and reported their performance below. These results further reinforce our key finding: even top-tier VLMs struggle with faithfully extracting structured data from scientific charts. Stronger reasoning capabilities alone do not yield better reconstruction precision since accurate data-point extraction remains a critical bottleneck.
>
> | Model            | Plot Precision | Table Accuracy |
> |------------------|----------------|----------------|
> | GPT-5            | 0.42          | 0.36          |
> | o4-mini          | 0.44        | 0.49          |
>
> [1] Brodeur, Abel & Mikola, Derek & Cook, Nikolai, 2024. "Mass Reproducibility and Replicability: A New Hope," IZA Discussion Papers 16912, Institute of Labor Economics (IZA).
>
> Regards,
>
> Paper 14065 Authors

---

### Note · Authors · 2026-01-04

**Comment:**

Dear AC and Reviewers,

Thank you very much for your thoughtful comments and constructive feedback. We have carefully addressed the suggestions and believe the paper has been substantially strengthened. After consideration, we have decided to withdraw this submission from the current review cycle.

We sincerely appreciate the time and effort you devoted to reviewing our work.

Regards,

Paper 14065 Authors

**Withdrawal Confirmation:**

I have read and agree with the venue's withdrawal policy on behalf of myself and my co-authors.